# BUSTLE: BOTTOM-UP PROGRAM SYNTHESIS THROUGH LEARNING-GUIDED EXPLORATION

**Augustus Odena,**[*] **Kensen Shi,**[*] **David Bieber, Rishabh Singh, Charles Sutton & Hanjun Dai**
Google Research
{augustusodena,kshi,dbieber,rising,charlessutton,hadai}@google.com

## ABSTRACT

Program synthesis is challenging largely because of the difficulty of search in a large space of programs. Human programmers routinely tackle the task of writing complex programs by writing sub-programs and then analyzing their intermediate results to compose them in appropriate ways. Motivated by this intuition, we present a new synthesis approach that leverages learning to guide a bottom-up search over programs. In particular, we train a model to prioritize compositions of intermediate values during search conditioned on a given set of input-output examples. This is a powerful combination because of several emergent properties. First, in bottom-up search, intermediate programs can be executed, providing semantic information to the neural network. Second, given the concrete values from those executions, we can exploit rich features based on recent work on property signatures. Finally, bottom-up search allows the system substantial flexibility in what order to generate the solution, allowing the synthesizer to build up a program from multiple smaller sub-programs. Overall, our empirical evaluation finds that the combination of learning and bottom-up search is remarkably effective, even with simple supervised learning approaches. We demonstrate the effectiveness of our technique on two datasets, one from the SyGuS competition and one of our own creation.

## 1 INTRODUCTION

Program synthesis is a longstanding goal of artificial intelligence research (Manna & Waldinger, 1971; Summers, 1977), but it remains difficult in part because of the challenges of search (Alur et al., 2013; Gulwani et al., 2017). The objective in program synthesis is to automatically write a program given a specification of its intended behavior, and current state of the art methods typically perform some form of search over a space of possible programs. Many different search methods have been explored in the literature, both with and without learning. These include search within a version-space algebra (Gulwani, 2011), bottom-up enumerative search (Udupa et al., 2013), stochastic search (Schkufza et al., 2013), genetic programming (Koza, 1994), reducing the synthesis problem to logical satisfiability (Solar-Lezama et al., 2006), beam search with a sequence-to-sequence neural network (Devlin et al., 2017), learning to perform premise selection to guide search (Balog et al., 2017), learning to prioritize grammar rules within top-down search (Lee et al., 2018), and learned search based on partial executions (Ellis et al., 2019; Zohar & Wolf, 2018; Chen et al., 2019).

While these approaches have yielded significant progress, none of them completely capture the following important intuition: human programmers routinely write complex programs by first writing sub-programs and then analyzing their intermediate results to compose them in appropriate ways. We propose a new learning-guided system for synthesis, called BUSTLE,[1] which follows this intuition in a straightforward manner. Given a specification of a program's intended behavior (in this paper given by input-output examples), BUSTLE performs bottom-up enumerative search for a satisfying program, following Udupa et al. (2013). Each program explored during the bottom-up search is an expression that can be executed on the inputs, so we apply a machine learning model to the resulting value to guide the search. The model is simply a classifier trained to predict whether the intermediate value produced by a partial program is part of an eventual solution. This combination of learning and

---

[*]Equal Contribution
[1]**B**ottom-**U**p program **S**ynthesis **T**hrough **L**earning-guided **E**xploration

bottom-up search has several key advantages. First, because the input to the model is a value produced by executing a partial program, the model's predictions can depend on semantic information about the program. Second, because the search is bottom-up, compared to previous work on execution-guided synthesis, the search procedure has more flexibility about which order to generate the program in, and this flexibility can be exploited by machine learning.

A fundamental challenge in this approach is that exponentially many intermediate programs are explored during search, so the model needs to be both fast and accurate to yield wall-clock time speedups. We are allowed to incur some slowdown from performing model inference, because if the model is accurate enough, we can search many fewer values before finding solutions. However, in the domains we consider, executing a program is still orders of magnitude faster than performing inference on even a small machine learning model, so this challenge cannot be ignored. We employ two techniques to deal with this. First, we arrange both the synthesizer and the model so that we can batch model prediction across hundreds of intermediate values. Second, we process intermediate expressions using property signatures (Odena & Sutton, 2020), which featurize program inputs and outputs using another set of programs.

A second challenge is that neural networks require large amounts of data to train, but there is no available data source of intermediate expressions. We can generate programs at random to train the model following previous work (Balog et al., 2017; Devlin et al., 2017), but models trained on random programs do not always transfer to human-written benchmarks (Shin et al., 2019). We show that our use of property signatures helps with this distribution mismatch problem as well.

In summary, this paper makes the following contributions:

- We present BUSTLE, which integrates machine learning into bottom-up program synthesis.
- We show how to efficiently add machine learning in the synthesis loop using property signatures and batched predictions. With these techniques, adding the model to the synthesizer provides an end-to-end improvement in synthesis time.
- We evaluate BUSTLE on two string transformation datasets: one of our own design and one from the SyGuS competition. We show that BUSTLE leads to improvements in synthesis time compared to a baseline synthesizer without learning, a DeepCoder-style synthesizer (Balog et al., 2017), and an encoder-decoder model (Devlin et al., 2017). Even though our model is trained on random programs, we show that its performance transfers to a set of human-written synthesis benchmarks.

## 2 BACKGROUND AND SETUP

### 2.1 PROGRAMMING BY EXAMPLE

In a Programming-by-Example (PBE) task (Winston, 1970; Menon et al., 2013; Gulwani, 2011), we are given a set of input-output pairs and the goal is to find a program such that for each pair, the synthesized program generates the corresponding output when executed on the input. To restrict the search space, the programs are typically restricted to a small domain-specific language (DSL). As an example PBE specification, consider the "io_pairs" given in Listing 1.

### 2.2 OUR STRING TRANSFORMATION DSL

Following previous work (Gulwani, 2011; Devlin et al., 2017), we deal with string and number transformations commonly used in spreadsheets. Such transformations sit at a nice point on the complexity spectrum as a benchmark task; they are simpler than programs in general purpose languages, but still expressive enough for many common string transformation tasks.

The domain-specific language we use (shown in Figure 1) is broadly similar to those of Parisotto et al. (2017) and Devlin et al. (2017), but compared to these, our DSL is expanded in several ways that make the synthesis task more difficult. First, in addition to string manipulation, our DSL includes integers, integer arithmetic, booleans, and conditionals. Second, our DSL allows for arbitrarily nested expressions, rather than having a maximum size. Finally, and most importantly, previous works (Gulwani, 2011; Devlin et al., 2017) impose a restriction of having `Concat` as the top-level operation. With this constraint, such approaches use version space algebras or dynamic programming

$$\begin{aligned}
\text{Expression } E &:= S \mid I \mid B \\
\text{String expression } S &:= \texttt{Concat}(S_1, S_2) \mid \texttt{Left}(S, I) \mid \texttt{Right}(S, I) \mid \texttt{Substr}(S, I_1, I_2) \\
&\quad \mid \texttt{Replace}(S_1, I_1, I_2, S_2) \mid \texttt{Trim}(S) \mid \texttt{Repeat}(S, I) \mid \texttt{Substitute}(S_1, S_2, S_3) \\
&\quad \mid \texttt{Substitute}(S_1, S_2, S_3, I) \mid \texttt{ToText}(I) \mid \texttt{LowerCase}(S) \mid \texttt{UpperCase}(S) \\
&\quad \mid \texttt{ProperCase}(S) \mid T \mid X \mid \texttt{If}(B, S_1, S_2) \\
\text{Integer expression } I &:= I_1 + I_2 \mid I_1 - I_2 \mid \texttt{Find}(S_1, S_2) \mid \texttt{Find}(S_1, S_2, I) \mid \texttt{Len}(S) \mid J \\
\text{Boolean expression } B &:= \texttt{Equals}(S_1, S_2) \mid \texttt{GreaterThan}(I_1, I_2) \mid \texttt{GreaterThanOrEqualTo}(I_1, I_2) \\
\text{String constants } T &:= \texttt{""} \mid \texttt{" "} \mid \texttt{","} \mid \texttt{"."} \mid \texttt{"!"} \mid \texttt{"?"} \mid \texttt{"("} \mid \texttt{")"} \mid \texttt{"["} \mid \texttt{"]"} \mid \texttt{"<"} \mid \texttt{">"} \\
&\quad \mid \texttt{"\{"} \mid \texttt{"\}"} \mid \texttt{"-"} \mid \texttt{"+"} \mid \texttt{"\_"} \mid \texttt{"/"} \mid \texttt{"\$"} \mid \texttt{"\#"} \mid \texttt{":"} \mid \texttt{";"} \mid \texttt{"@"} \mid \texttt{"\%"} \mid \texttt{"0"} \\
&\quad \mid \text{string constants extracted from I/O examples} \\
\text{Integer constants } J &:= 0 \mid 1 \mid 2 \mid 3 \mid 99 \\
\text{Input } X &:= x_1 \mid \ldots \mid x_k
\end{aligned}$$

Figure 1: Domain-specific language (DSL) of expressions considered in this paper.

to exploit the property that partial programs must form substrings of the output. Our DSL lifts this constraint, allowing the synthesizer to handle more kinds of tasks than in those previous works.

Our DSL allows for compositions of common string transformation functions. These functions include string concatenation (`Concat`) and other familiar string operations (listed in Figure 1 and discussed further in Appendix A). Integer functions include arithmetic, finding the index of substrings (`Find`), and string length. Finally, commonly useful string and integer constants are included. We also use heuristics to extract string constants that appear multiple times in the input-output examples.

## 2.3 BOTTOM-UP SYNTHESIS

The baseline synthesizer on top of which we build our approach is a bottom-up enumerative search inspired by Udupa et al. (2013), which enumerates DSL expressions from smallest to largest, following Algorithm 1 if the lines colored in blue (4, 16, and 17) are removed. This baseline uses a *value-based* search. During the search each candidate expression is executed to see if it meets the specification. Then, rather than storing the expressions that have been produced in the search, we store the values produced by executing the expressions. This allows the search to avoid separately extending sub-expressions that are semantically equivalent on the given examples. If there are $n$ separate input-output examples for a given task, each value represents one code expression and internally contains the results of executing that expression on inputs from all $n$ examples. Hence, two values that represent different code expressions are semantically equivalent on the examples if their $n$ contained results all match each other.

Every expression has an integer weight, which for the baseline synthesizer is the number of nodes in the abstract syntax tree (AST). The search maintains a table mapping weights to a list of all the values of previously explored sub-expressions of that weight. The search is initialized with the set of input variables as well as any constants extracted with heuristics, all of which have weight 1. Extracted constants include common symbols and delimiters that appear at least once, and long substrings that appear multiple times in the example strings. The search then proceeds to create all expressions of weight 2, and then of weight 3, and so on. To create all values of a particular weight, we loop over all available functions, calling each function with all combinations of arguments that would yield the desired weight. For example, if we are trying to construct all values of weight 10 of the form `Concat`$(x, y)$, we iterate over all values where $x$ has weight 1 and $y$ has weight 8, and then where $x$ has weight 2 and $y$ has weight 7, and so forth. (The `Concat` function itself contributes weight 1.) Every time a new expression is constructed, we evaluate it on the given inputs, terminating the search when the expression produces the desired outputs.

```python
1  io_pairs = [
2    ("butter",  "butterfly"),
3    ("abc",     "abc_"),
4    ("xyz",     "XYZ_"),
5  ]
6
7  p1 = lambda inp, outp: inp in outp
8  p2 = lambda inp, outp: outp.endswith(inp)
9  p3 = lambda inp, outp: inp.lower() in outp.lower()
```

Listing 1: An example set of input-output pairs, along with three properties that can act on them. The first returns True for the first two pairs and False for the third. The second returns False for all pairs. The third returns True for all pairs. The resulting property signature is {Mixed, AllFalse, AllTrue}. These examples are written in Python for clarity, but our implementation is in Java.

---

**Algorithm 1** The BUSTLE Synthesis Algorithm

---

**Input:** Input-output examples $(\mathcal{I}, \mathcal{O})$
**Output:** A program $P$ consistent with the examples $(\mathcal{I}, \mathcal{O})$
**Auxiliary Data:** Supported operations *Ops*, supported properties *Props*, and a model $M$ trained using *Props* as described in Section 3.1.

1: $E \leftarrow \emptyset$       ▷ $E$ maps integer weights to terms with that weight
2: $C \leftarrow \text{EXTRACTCONSTANTS}(\mathcal{I}, \mathcal{O})$
3: $E[1] \leftarrow \mathcal{I} \cup C$       ▷ Inputs and constants have weight 1
4: $s_{io} \leftarrow \text{PROPERTYSIGNATURE}(\mathcal{I}, \mathcal{O}, Props)$
5: **for** $w = 2, \ldots$ **do**       ▷ Loop over all possible term weights
6:    **for all** $op \in Ops$ **do**
7:      $n \leftarrow op.arity$
8:      $A \leftarrow \emptyset$       ▷ $A$ holds all argument tuples
9:      **for all** $[w_1, \ldots, w_n]$       ▷ Make all arg tuples with these weights that type-check
10:          s.t. $\sum_i w_i = w - 1, \ w_i \in \mathbb{Z}^+$ **do**
11:       $A \leftarrow A \cup \{(a_1, \ldots, a_n) \mid a_i.weight = w_i \wedge a_i.type = op.argtypes_i\}$
12:      **for all** $(a_1, \ldots, a_n) \in A$ **do**
13:        $\mathcal{V} \leftarrow \text{EXECUTE}(op, (a_1, \ldots, a_n))$
14:        **if** $\mathcal{V} \notin E$ **then**       ▷ The value has not been encountered before
15:          $w' \leftarrow w$
16:          $s_{vo} \leftarrow \text{PROPERTYSIGNATURE}(\mathcal{V}, \mathcal{O}, Props)$
17:          $w' \leftarrow \text{REWEIGHTWITHMODEL}(M, s_{io}, s_{vo}, w)$
18:          $E[w'] \leftarrow E[w'] \cup \{\mathcal{V}\}$
19:        **if** $\mathcal{V} = \mathcal{O}$ **then**
20:          **return** EXPRESSION($\mathcal{V}$)

---

## 2.4 PROPERTY SIGNATURES

In order to perform machine learning on values encountered during the enumeration, we make use of recent work on property signatures (Odena & Sutton, 2020). Consider a function with input type $\tau_{in}$ and output type $\tau_{out}$. In this context, a property is a function of type: $(\tau_{in}, \tau_{out}) \rightarrow \texttt{Bool}$ that describes some aspect of the function under consideration. If we have a list of such properties and some inputs and outputs of the correct type, we can evaluate all the properties on the input-output pairs to get a list of outputs that we will call the property signature. More precisely, given a list of $n$ input-output pairs and a list of $k$ properties, the property signature is a length $k$ vector. The elements of the vector corresponding to a given property will have one of the values AllTrue, AllFalse, and Mixed, depending on whether the property returned True for all $n$ pairs, False for all $n$ pairs, or True for some and False for others, respectively. Concretely, then, any property signature can be identified with a trit-vector, and we represent them in computer programs as arrays containing the values $\{-1, 0, 1\}$. An example of property signatures is shown in Listing 1.

```
1  # Compute the depth of a path, i.e., count the number of slashes
2  solution = "TO_TEXT(MINUS(LEN(var_0), LEN(SUBSTITUTE(var_0, \"/\", \"\"))))"
3  io_pairs = {"/this/is/a/path": "4", "/home": "1", "/a/b": "2"}
4
5  # Change DDMMYYYY date to MM/DD/YYYY
6  solution = "CONCATENATE(MID(var_0, 3, 2), \"/\", REPLACE(var_0, 3, 2, \"/\"))"
7  io_pairs = {"08092019": "09/08/2019", "12032020": "03/12/2020"}
8
9  # Create capitalized acronym from two words in one cell
10 solution =
11     "UPPER(CONCATENATE(LEFT(var_0, 1), MID(var_0, ADD(FIND(\" \", var_0), 1), 1)))"
12 io_pairs = {"product area": "PA", "Vice president": "VP"}
```

Listing 2: Three of our benchmark problems (all solved by BUSTLE).

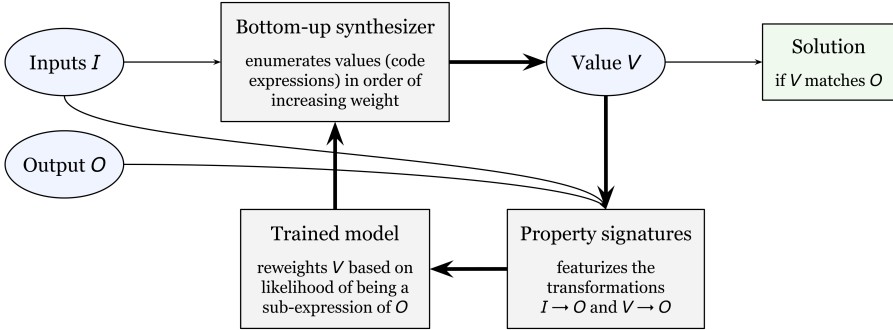

Figure 2: Diagram outlining the BUSTLE approach. The bold arrows show the main feedback loop. The bottom-up synthesizer enumerates values (code expressions), which are featurized along with the I/O example using property signatures. The property signatures are passed to a trained model that reweights the value based on whether it appears to be a sub-expression of a solution, and the reweighted value is given back to the bottom-up synthesizer for use in enumerating larger expressions.

## 2.5 BENCHMARKS

We evaluate BUSTLE on two datasets. The first dataset is a new suite of 38 human-written benchmark tasks, which were designed to contain a variety of tasks difficult enough to stress our system. Some tasks involve conditionals, which are not present in our other set of benchmark tasks (from SyGuS). The search space explored by the synthesizers to solve these tasks is quite large: on average, BUSTLE searches about 5 million expressions per benchmark attempt, and 1 million expressions per successful attempt. Most tasks have between 2 and 4 input-output pairs, though for some tasks, more than 4 pairs are needed to fully specify the semantics of the desired program, especially when conditionals are involved. In each case, we gave what we felt was the number of pairs a user of such a system would find reasonable (though of course this is a subjective judgment). Three representative benchmarks are shown in Listing 2. See Appendix B for a full list.

The second dataset consists of all SyGuS programs from the 2019 PBE SLIA TRACK and the 2018 PBE Strings Track whose inputs and outputs are only strings. We removed duplicate copies of problems which simply had extra examples. This results in 89 remaining tasks.

## 3 BUSTLE: BOTTOM-UP SYNTHESIS WITH LEARNING

BUSTLE uses a trained machine learning model to guide the bottom-up search, outlined in Figure 2. Suppose the synthesis task includes $n$ different input-output examples, and there are $m$ separate input variables (such that the function to synthesize has arity $m$). Recall that every value in the bottom-up search represents a code expression and also contains the $n$ results of evaluating the code expression on the $n$ examples. To guide the bottom-up search, we run a machine learning model on the results contained in intermediate values encountered during the search.

The model is a binary classifier $p(y \mid \mathcal{I}, \mathcal{V}, \mathcal{O})$ where $\mathcal{I}$ is a set of $m$ input values (one for each input variable), $\mathcal{O}$ is a single value representing the desired output, and $\mathcal{V}$ is an intermediate value encountered during the bottom-up search. We want the model to predict the binary label $y = 1$ if the code expression that produced $\mathcal{V}$ is a sub-expression of a solution to the synthesis problem, and $y = 0$ otherwise. Given these predictions, we de-prioritize sub-expressions that are unlikely to appear in the final result, which, when done correctly, dramatically speeds up the synthesizer.

## 3.1 Model Architecture and Training

Because we want the classifier to learn whether a value is intermediate between an input and an output, the model is conditioned on *two* property signatures: one from the inputs to the output, and one from the intermediate value to the output. Recall from Section 2.4 that a property signature is computed by applying a list of properties to a list of input-output pairs. Thus, one signature is computed by applying all of the properties to input-output pairs, and the other is applied to intermediate value-output pairs. A few example properties that we use include: (a) if the value $v$ and output $o$ are both strings, is $v$ an initial substring of $o$; (b) do $v$ and $o$ have the same length; (c) does the string contain a space, and so on. (See Appendix C for the full list of properties.) Then we concatenate these two vectors to obtain the model's input. The rest of the model is straightforward. Each element of the property signature is either AllTrue, AllFalse, or Mixed. We embed the ternary property signature into a higher-dimensional dense vector and then feed it into a fully connected neural network for binary classification.

This model is simple, but we are only able to use such a simple model due to our particular design choices: our form of bottom-up search guarantees that all intermediate expressions can yield a value comparable to the inputs and outputs, and property signatures can do much of the "representational work" that would otherwise require a larger or more complex model.

The classifier is trained by behavior cloning on a set of training problems. However, obtaining a training dataset is challenging. Ideally, the training set would contain synthesis tasks that are interesting to humans, but such datasets can be small compared to what is needed to train deep neural networks. Instead, we train on randomly generated synthetic data, similar to Devlin et al. (2017). This choice does come with a risk of poor performance on human-written tasks due to domain mismatch (Shin et al., 2019), but we show in Section 4 that BUSTLE can overcome this issue.

Generating the synthetic data is itself nontrivial. Because different DSL functions have corresponding argument preconditions and invariants (e.g., several functions take integer indices which must be in range for a given string), a random sampling of DSL programs and inputs would lead to a large number of training examples where the program cannot be applied to the sampled inputs.

Instead, we use the idea of generating data from synthesis searches, as in TF-Coder (Shi et al., 2020). First, we generate some random input strings to form the inputs $\mathcal{I}$ (using multiple input variables and multiple examples). From these inputs, we run bottom-up search using a dummy output, so that the search will keep generating expressions. We randomly select some generated expressions and pretend that they are outputs $\mathcal{O}$ for a synthesis task. Then, for each selected expression, we randomly select one of its sub-expressions $\mathcal{V}$ to form a positive training example $(\mathcal{I}, \mathcal{V}, \mathcal{O})$. We also randomly select another expression $\mathcal{V}'$ from the search that is *not* a sub-expression of $\mathcal{O}$, to serve as a negative training example $(\mathcal{I}, \mathcal{V}', \mathcal{O})$. In our experiments, we perform 1000 searches on random inputs, select 100 values at random from each search to serve as outputs, and create one positive and one negative example for each selected value as described. Considering that a dataset of size 200,000 is not large by modern deep learning standards, training the model is a small one-time cost, completing in a few hours on CPU.

## 3.2 Combining Model with Synthesis

Incorporating the model into bottom-up synthesis is straightforward, and can be accomplished by adding the blue lines into Algorithm 1. Lines 4 and 16 compute the property signatures required for the model input, as described previously. The main challenge is that the model produces a probability $p(y \mid \mathcal{I}, \mathcal{V}, \mathcal{O})$, but the search is organized by integer weights. We resolve this with a simple heuristic: at the time we generate a new value, we have the weight $w$ of the expression that generates it. We discretize the model's output probability into an integer in $\delta \in \{0, \ldots, 5\}$ by binning it into six bins

bounded by $[0.0, 0.1, 0.2, 0.3, 0.4, 0.6, 1.0]$. The new weight is computed from the discretized model output as $w' = w + 5 - \delta$. This function is indicated by REWEIGHTWITHMODEL in Algorithm 1.

A key challenge is making the model fast enough. Evaluating the model once for every intermediate value could cause the synthesizer to slow down so much that the overall performance would be worse than with no model at all. However, BUSTLE actually outperforms our baselines even when measured strictly in terms of wall-clock time (Section 4). There are several reasons for this. First, computing property signatures for the expressions allows us to take some of the work of representing the intermediate state out of the neural network and to do it natively in Java (which is much faster comparatively). Second, because a property signature is a fixed-length representation, it can be fed into a simple feed-forward neural network, rather than requiring a recurrent model, as would be necessary if we passed in a more complex representation such as the AST. Third, because of this fixed-length representation, it is easy to batch many calls to the machine learning model and process them using CPU vector instructions. Inference calls to the machine learning model could, in principle, be done in parallel to the rest of the synthesizer, either on a separate CPU core or on an accelerator, which would further improve wall-clock results, but our experiments are performed entirely on one CPU. Due to these optimizations, computing property signatures and running the model on them accounts for only roughly 20% of the total time spent.

## 4  EXPERIMENTS

We evaluate BUSTLE on both datasets described in Section 2.5. To measure performance, we consider the number of benchmarks solved as a function of the number of candidates considered, which gives insight into how well the model can guide the search. We additionally consider benchmarks solved as a function of wall-clock time, which takes into account the computational cost of model inference.

We compare BUSTLE to five other methods:

1. A baseline bottom-up synthesizer without machine learning, which explores expressions in order of increasing size, without any model to guide the search toward the desired output.
2. The baseline synthesizer augmented with domain-specific heuristics (substring relationships and edit distance) to reweight intermediate string values during the search.
3. An encoder-decoder model as in RobustFill (Devlin et al., 2017), which predicts a program directly from the input-output examples. We use beam search on the decoder with a beam size of $80,000$ programs, enough to exhaust 16 GB of GPU memory. See Appendix D for more details.
4. A premise selection model as in DeepCoder (Balog et al., 2017), which lets us analyze whether learning within the synthesis loop is better than learning once at the beginning of the search. We train a model similar to the model trained for BUSTLE on the same dataset, but instead of predicting whether an expression is a sub-expression of a solution, we predict which operations will be used in a solution. The examples are given to the model using character-level embeddings. Then, for each benchmark, we exclude the 2 operations that the model predicts are the least likely.
5. A premise selection model that uses property signatures instead of character-level embeddings.

**Results**  The results on our 38 new benchmarks are shown in Figure 3a. Whether comparing by the number of expressions (left) or the wall-clock time (right), BUSTLE (red-dash) performs quite well, solving 31 tasks within 30 million candidate expressions or 30 seconds. It outperforms all other methods besides one that uses *both* the model and the heuristics (purple). In particular, BUSTLE outperforms the domain-specific heuristics, even though the heuristics are much faster to execute compared to running model inference. Furthermore, when using both the model and heuristics (purple), the synthesizer performs the best overall. This indicates that although learning in the loop outperforms domain-specific heuristics, the two approaches can be combined to achieve better performance than either alone.

Results on SyGuS benchmarks in Figure 3b were broadly similar, with one exception: the handwritten heuristics perform slightly better than (heuristic-free) BUSTLE. There are a few important caveats to this, however. First, the SyGuS problems are slightly different from the kinds of problems in our 38 new benchmark tasks and the training data, e.g., some SyGuS problems have an incredibly large number of examples or use string-manipulation functionality outside our DSL. Second, our heuristics substantially outperform all other baselines, which suggests that they are strong heuristics to begin

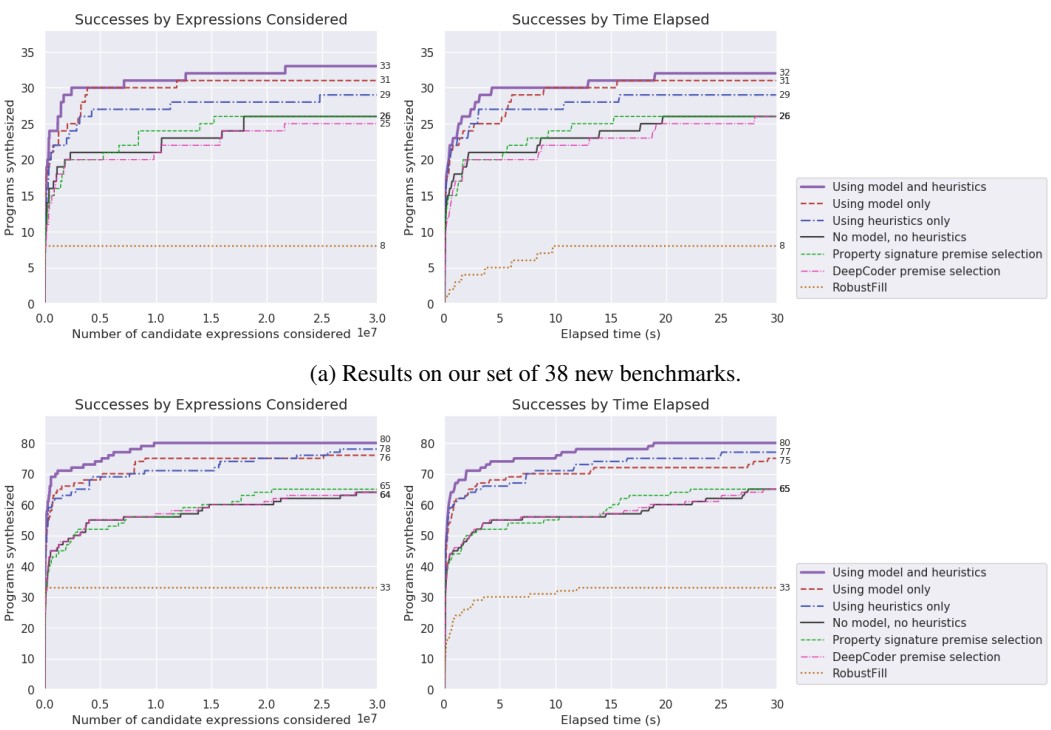

(a) Results on our set of 38 new benchmarks.

(b) Results on 89 benchmarks from SyGuS.

Figure 3: (Left) Benchmarks solved as a function of intermediate expressions considered. This metric makes BUSTLE look somewhat better than it is, because it ignores slowdowns in wall-clock time, but it is still important to analyze. It is invariant to engineering considerations, providing an upper bound on how well we can do in wall-clock terms through speeding up the model. (Right) Benchmarks solved over elapsed wall-clock time. BUSTLE still outperforms all baselines on our 38 new tasks, but not by quite as much due to time spent on model inference.

with. Third, we still see a substantial improvement by *combining* the BUSTLE model with heuristics, so the best performing algorithm does indeed use BUSTLE.

BUSTLE outperforms both DeepCoder-style premise selection methods (green and pink). Premise selection allows some tasks to be solved faster, but it does not lead to more tasks being solved overall compared to the no-model baseline (black). This is evidence that learning in the loop is important to guide the search as it happens, and one step of learning at the beginning of search is not as effective. We furthermore observe that using property signatures (green) leads to better performance than not (pink), since they can help the models be more robust to train-test distribution shift.

BUSTLE also outperforms RobustFill (orange). Relatively speaking, RobustFill performs better on the SyGuS tasks than our new tasks, which may indicate that some of our new tasks are more difficult due to the use of conditionals. Overall, RobustFill does not perform well, possibly because the end-to-end neural approach is less robust to the train-test distribution shift, and because its complex model cannot predict programs as quickly as BUSTLE's fast combinatorial search.

We conduct two additional analyses to better understand the performance of BUSTLE. First, we investigate the predictions of the model when it is run on the intermediate values encountered during synthesis of the human-written benchmarks. We generate histograms for the model's predictions on expressions that do appear in the solution and expressions that do not appear in the solution, for all benchmarks that were solved. Predictions for true sub-expressions skew positive and predictions for negative sub-expressions skew negative. This provides further evidence that our model generalizes well to human benchmarks, despite the domain mismatch to the synthetic data used in training. The full results are in Appendix E. Finally, we determined that all but one of the benchmarks solved by the baseline (no model, no heuristics) were also solved by BUSTLE, across both sets of benchmarks.

## 5 RELATED WORK

For surveys on program synthesis and machine learning for software engineering, see Gottschlich et al. (2018); Solar-Lezama (2018); Gulwani et al. (2017); Allamanis et al. (2018). A well-known synthesizer for spreadsheet programs is FlashFill (Gulwani, 2011), based on Version Space Algebras (VSAs) which are powerful and fast. However, VSAs only apply to restricted DSLs, e.g., the top-most function in the program must be `Concat`, which allows it to perform efficient divide and conquer style search. Our technique has no such restrictions.

Early work on machine learning for program synthesis includes DeepCoder (Balog et al., 2017), which uses a learned model to select once at the beginning of search. Although this idea is pragmatic, the disadvantage is that once the search has started, the model can give no further feedback. Odena & Sutton (2020) use property signatures within a DeepCoder-style model for premise selection.

One can also train a machine learning model to emit whole programs token-by-token using an encoder-decoder neural architecture (Bunel et al., 2018; Devlin et al., 2017; Parisotto et al., 2017), but this approach does not have the ability to inspect outputs of intermediate programs. Previous work has also considered using learning within syntax guided search over programs (Yin & Neubig, 2017; Lee et al., 2018), but because these methods are top-down, it is much more difficult to guide them by execution information, since partial programs will have holes. Finally, Nye et al. (2019) learns to emit a partial program and fill in the holes with a symbolic search.

The most closely related work to ours is *neural execution-guided synthesis*, which like BUSTLE uses values produced by intermediate programs within a neural network. Zohar & Wolf (2018) process intermediate values of a program using a neural network for a small, straight-line DSL, but they do not use the model to evaluate intermediate programs. Another approach is to rewrite a programming language so that it can be evaluated "left-to-right", allowing values to be used to prioritize the search in an actor-critic framework (Ellis et al., 2019). Similarly, Chen et al. (2019) use intermediate values while synthesizing a program using a neural encoder-decoder model, but again this work proceeds in a variant of left-to-right search that is modified to handle conditionals and loops. None of these approaches exploit our main insight, which is that bottom-up search allows the model to prioritize and combine small programs that solve different subtasks.

*Learning to search* has been an active area in machine learning, especially in imitation learning (Daumé et al., 2009; Ross et al., 2011; Chang et al., 2015). Combining more sophisticated imitation learning strategies into BUSTLE is an interesting direction for future work.

## 6 CONCLUSION

We introduced BUSTLE, a technique for using machine learning to guide bottom-up search for program synthesis. BUSTLE exploits the fact that bottom-up search makes it easy to evaluate partial programs, and it uses machine learning to predict the likelihood that a given intermediate value is a sub-expression of the desired solution. We have shown that BUSTLE improves over various baselines, including recent deep-learning-based program synthesis approaches (DeepCoder and RobustFill), on two challenging benchmark suites of string-manipulation tasks, in terms of candidate programs considered as well as wall-clock time. In fact, showing that learning-in-the-loop can be made fast enough for program synthesis is perhaps the major contribution of this work. The idea of learning-in-the-loop, though novel as far as we are aware, is relatively obvious, but through this work we learned that it can be efficient enough to provide time speedups overall.

### ACKNOWLEDGMENTS

The authors thank Sam Schoenholz and the anonymous conference reviewers for their helpful reviews and comments on our paper.

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

## A    EXPANDED DESCRIPTION OF DSL

Our DSL allows for nesting and compositions of common string transformation functions. These functions include string concatenation (`Concat`); returning a substring at the beginning (`Left`), middle (`Substr`), or right (`Right`) of a string; replacing a substring of one string, indicated by start and end position, with another string (`Replace`); removing white space from the beginning and ending of a string (`Trim`); concatenating a string with itself a specified number of times (`Repeat`); substituting the first $k$ occurrences of a substring with another (`Substitute`); converting an integer to a string (`ToText`); and converting a string to `LowerCase`, `UpperCase`, or every word capitalized (`ProperCase`). Integer functions include arithmetic, returning the index of the first occurrence of a substring within a string (`Find`), and string length (`Len`). We also have some functions either consuming or producing booleans (`If`, `Equals`, `GreaterThan`, `GreaterThanOrEqualTo`). Finally, a few commonly useful string and integer constants are included.

## B    LIST OF BENCHMARK PROGRAMS

Here we show each of our 38 human-written benchmark problems, and a possible solution written in a DSL that is a superset of the DSL used by our synthesizer. We have separated them into Listing 3 Listing 4 for space reasons. Note that the synthesizer can and does solve problems with programs different than the programs given here, and that it does not solve all of the problems.

## C    LISTING OF PROPERTIES USED

BUSTLE computes two types of property signatures: the signature involving the inputs and the outputs, and the signature involving the intermediate state and the outputs. In this paper, the inputs and outputs are always strings, but the intermediate state may be an integer, a string, or a boolean. In an abstract sense, a property acts on an input and an output, but some properties will simply ignore the input, and so we implement those as functions with only one argument. Thus, we have six types of properties in principle:

- properties acting on a single string (Listing 5).
- properties acting on a single integer (Listing 6).
- properties acting on a single boolean (there is only one of these).
- properties acting on a string and the output string (Listing 7).
- properties acting on an integer and the output string (Listing 8).
- properties acting on a boolean and the output string (we don't actually use any of these presently).

For a program with multiple inputs, we simply loop over all the inputs and the output, computing all of the relevant types of properties for each. For example, a program taking two string inputs and yielding one string output will have single-argument string properties for the output and two sets of double-argument string properties, one for each input. We fix a maximum number of inputs and pad the signatures so that they are all the same size.

## D    MORE IMPLEMENTATION DETAILS ABOUT ROBUSTFILL BASELINE

To make the vanilla model proposed in Devlin et al. (2017) work on our benchmarks, we have made the following necessary changes:

- As the vanilla model only allows single input per each input-output pair, we here concatenate the variable number of inputs with a special separator token.
- Following Devlin et al. (2017), the vocabulary of input-output examples are constructed at the character level, as there are numerous integer and string literals in our benchmark and training tasks.
- The desired programs may use string constants which may depend on the particular input-output examples. In addition to predicting program tokens, RobustFill may also need to predict the string constants character-by-character when necessary.

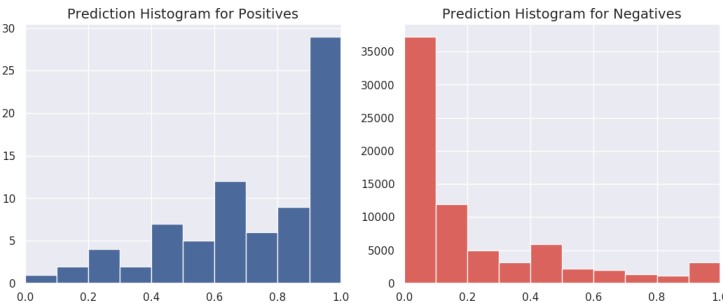

Figure 4: Histograms of model predictions for expressions seen while solving benchmarks. (Left) for expressions that were sub-expressions of a solution, the majority received predictions close to 1, showing that the model can identify the correct expressions to prioritize during search. (Right) for expressions that were not sub-expressions of a solution, predictions skewed close to 0.

- There could be out-of-vocabulary characters in the input-output examples, as the test benchmarks have a different data distribution than synthetic training examples. In this case, we replace these characters with spaces.

We use the same training data to train the RobustFill model. We first retain 10% of the training samples for validation, and identify the best number of training epochs. Then we retrain the model using full data for that many epochs. The batch size we use is 1024, with fixed learning rate 1e-3. When decoding the program, we use a 3-layer LSTM with embedding size of 512.

During inference, we use beam search to obtain the most likely $M$ programs, following Devlin et al. (2017). On a single GPU with 16 GB of memory, the maximum beam size we can use is $M = 80,000$, which is already several magnitudes larger than $1000$ (the largest beam size used in original paper). It takes roughly 25 seconds to perform this beam search, which is conveniently close to the 30 second time limit in our experiments.

## E    ANALYSIS OF MODEL PREDICTIONS

We investigate the predictions of the model when it is run on the intermediate values actually encountered during synthesis of the human-written benchmarks. We compute separate histograms for the model's predictions on expressions that do appear in the solution and expressions that do not appear in the solution, for all benchmarks that were solved. Predictions for true sub-expressions skew positive and predictions for negative sub-expressions skew negative. This provides further evidence that our model generalizes well to human benchmarks, despite the domain mismatch to the synthetic data used in training. See Figure 4.

```
1  // add decimal point if not present
2  IF(ISERROR(FIND(".", var_0)), CONCATENATE(var_0, ".0"), var_0)
3
4  // add plus sign to positive integers
5  IF(EXACT(LEFT(var_0, 1), "-"), var_0, CONCATENATE("+", var_0))
6
7  // append AM or PM to the hour depending on if it's morning
8  CONCATENATE(LEFT(var_0, MINUS(FIND(":",var_0), 1)), IF(EXACT(var_1, "morning"), " AM", " PM"))
9
10 // fix capitalization of city and state
11 CONCATENATE(LEFT(PROPER(var_0), MINUS(LEN(var_0), 1)), UPPER(RIGHT(var_0, 1)))
12
13 // capitalize the first word and lowercase the rest
14 REPLACE(LOWER(var_0), 1, 1, UPPER(LEFT(var_0, 1)))
15
16 // whether the first string contains the second
17 TO_TEXT(ISNUMBER(FIND(var_1, var_0)))
18
19 // whether the first string contains the second, ignoring case
20 TO_TEXT(ISNUMBER(FIND(LOWER(var_1), LOWER(var_0))))
21
22 // count the number of times the second string appears in the first
23 TO_TEXT(DIVIDE(MINUS(LEN(var_0), LEN(SUBSTITUTE(var_0, var_1, ""))), LEN(var_1)))
24
25 // create email address from name and company
26 LOWER(CONCATENATE(LEFT(var_0, 1), var_1, "@", var_2, ".com"))
27
28 // change DDMMYYYY date to MM/DD/YYYY
29 CONCATENATE(MID(var_0, 3, 2), "/", REPLACE(var_0, 3, 2, "/"))
30
31 // change YYYY-MM-DD date to YYYY/MM/DD
32 SUBSTITUTE(var_0, "-", "/")
33
34 // change YYYY-MM-DD date to MM/DD
35 SUBSTITUTE(RIGHT(var_0, 5), "-", "/")
36
37 // extract the part of a URL between the 2nd and 3rd slash
38 MID(var_0, ADD(FIND("//", var_0), 2), MINUS(MINUS(FIND("/", var_0, 9), FIND("/", var_0)), 2))
39
40 // extract the part of a URL starting from the 3rd slash
41 RIGHT(var_0, ADD(1, MINUS(LEN(var_0), FIND("/", var_0, ADD(FIND("//", var_0), 2)))))
42
43 // get first name from second column
44 LEFT(var_1, MINUS(FIND(" ", var_1), 1))
45
46 // whether the string is lowercase
47 IF(EXACT(var_0, LOWER(var_0)), "true", "false")
48
49 // get last name from first column
50 RIGHT(var_0, MINUS(LEN(var_0), FIND(" ", var_0)))
51
52 // output "Completed" if 100%, "Not Yet Started" if 0%, "In Progress" otherwise
53 IF(var_0="100%", "Completed", IF(var_0="0%", "Not Yet Started", "In Progress"))
54
55 // enclose negative numbers in parentheses
56 IF(EXACT(LEFT(var_0, 1), "-"), CONCATENATE(SUBSTITUTE(var_0, "-", "("), ")"), var_0)
```

Listing 3: Potential solutions for our benchmarks, along with comments describing the semantics of the solution.

```
1  // pad text with spaces to a given width
2  CONCATENATE(REPT(" ", MINUS(VALUE(var_1), LEN(var_0))), var_0)
3
4  // pad number with 0 to width 5
5  CONCATENATE(REPT("0", MINUS(5, LEN(var_0))), var_0)
6
7  // the depth of a path, i.e., count the number of /
8  TO_TEXT(MINUS(LEN(var_0), LEN(SUBSTITUTE(var_0, "/", ""))))
9
10 // extract the rest of a word given a prefix
11 RIGHT(var_0, MINUS(LEN(var_0), LEN(var_1)))
12
13 // prepend Mr. to last name
14 CONCATENATE("Mr. ", RIGHT(var_0, MINUS(LEN(var_0), FIND(" ", var_0))))
15
16 // prepend Mr. or Ms. to last name depending on gender
17 CONCATENATE(IF(EXACT(var_1, "male"), "Mr. ", "Ms. "),
18             RIGHT(var_0, MINUS(LEN(var_0), FIND(" ", var_0))))
19
20 // remove leading and trailing spaces and tabs, and lowercase
21 TRIM(LOWER(var_0))
22
23 // replace <COMPANY> in a string with a given company name
24 SUBSTITUTE(var_0, "<COMPANY>", var_1)
25
26 // replace com with org
27 SUBSTITUTE(var_0, "com", "org", 1)
28
29 // select the first string, or the second if the first is NONE
30 IF(EXACT(var_0, "NONE"), var_1, var_0)
31
32 // select the longer of 2 strings, defaulting to the first if equal length
33 IF(GT(LEN(var_1), LEN(var_0)), var_1, var_0)
34
35 // whether the two strings are exactly equal, yes or no
36 IF(EXACT(var_0, var_1), "yes", "no")
37
38 // whether the two strings are exactly equal ignoring case, yes or no
39 IF(EXACT(LOWER(var_0), LOWER(var_1)), "yes", "no")
40
41 // length of string
42 TO_TEXT(LEN(var_0))
43
44 // extract the rest of a word given a suffix
45 LEFT(var_0, MINUS(LEN(var_0), LEN(var_1)))
46
47 // swap the case of a string that is entirely uppercase or lowercase
48 IF(EXACT(var_0, LOWER(var_0)), UPPER(var_0), LOWER(var_0))
49
50 // truncate and add ... if longer than 15 characters
51 IF(GT(LEN(var_0), 15), CONCATENATE(LEFT(var_0, 15), "..."), var_0)
52
53 // create acronym from two words in one cell
54 CONCATENATE(LEFT(var_0, 1), MID(var_0, ADD(FIND(" ", var_0), 1), 1))
55
56 // create capitalized acronym from two words in one cell
57 UPPER(CONCATENATE(LEFT(var_0, 1), MID(var_0, ADD(FIND(" ", var_0), 1), 1)))
```

Listing 4: Potential solutions for our benchmarks, along with comments describing the semantics of the solution.

```
1  str.isEmpty()                  // is empty?
2  str.length() == 1             // is single char?
3  str.length() <= 5             // is short string?
4  str.equals(lower)             // is lowercase?
5  str.equals(upper)             // is uppercase?
6  str.contains(" ")             // contains space?
7  str.contains(",")             // contains comma?
8  str.contains(".")             // contains period?
9  str.contains("-")             // contains dash?
10 str.contains("/")             // contains slash?
11 str.matches(".*\\d.*")        // contains digits?
12 str.matches("\\d+")           // only digits?
13 str.matches(".*[a-zA-Z].*")   // contains letters?
14 str.matches("[a-zA-Z]+")      // only letters?
```

Listing 5: Java code for all Properties acting on single Strings.

```
1  integer == 0                  // is zero?
2  integer == 1                  // is one?
3  integer == 2                  // is two?
4  integer < 0                   // is negative?
5  0 < integer && integer <= 3   // is small integer?
6  3 < integer && integer <= 9   // is medium integer?
7  9 < integer                   // is large integer?
```

Listing 6: Java code for all Properties acting on single Integers.

```
1  outputStr.contains(str)                // output contains input?
2  outputStr.startsWith(str)              // output starts with input?
3  outputStr.endsWith(str)                // output ends with input?
4  str.contains(outputStr)                // input contains output?
5  str.startsWith(outputStr)              // input starts with output?
6  str.endsWith(outputStr)                // input ends with output?
7  outputStrLower.contains(lower)         // output contains input ignoring case?
8  outputStrLower.startsWith(lower)       // output starts with input ignoring case?
9  outputStrLower.endsWith(lower)         // output ends with input ignoring case?
10 lower.contains(outputStrLower)         // input contains output ignoring case?
11 lower.startsWith(outputStrLower)       // input starts with output ignoring case?
12 lower.endsWith(outputStrLower)         // input ends with output ignoring case?
13 str.equals(outputStr)                  // input equals output?
14 lower.equals(outputStrLower)           // input equals output ignoring case?
15 str.length() == outputStr.length()     // input same length as output?
16 str.length() < outputStr.length()      // input shorter than output?
17 str.length() > outputStr.length()      // input longer than output?
```

Listing 7: Java code for all Properties acting on a String and the output String.

```
1  integer < outputStr.length()                   // is less than output length?
2  integer <= outputStr.length()                  // is less or equal to output length?
3  integer == outputStr.length()                  // is equal to output length?
4  integer >= outputStr.length()                  // is greater or equal to output length?
5  integer > outputStr.length()                   // is greater than output length?
6  Math.abs(integer - outputStr.length()) <= 1    // is very close to output length?
7  Math.abs(integer - outputStr.length()) <= 3    // is close to output length?
```

Listing 8: Java code for all Properties acting on an Integer and the output String.