# OpenReview forum: "BUSTLE: Bottom-Up Program Synthesis Through Learning-Guided Exploration"
_ICLR.cc/2021/Conference — ICLR 2021 Spotlight_

### Official Review · AnonReviewer4 · 2020-10-22
**Well written paper but questions about contribution significance. Possible reject.**

**Rating:** 5
**Confidence:** 3

**Review:**

The paper proposes to add machine learning component to the bottom-up program synthesis algorithm. Machine learning component uses information from property signatures to prioritize the candidate expressions for further expression space exploration. Authors show that their algorithm outperforms other approaches on the benchmark sets.

Positives:
- The paper presents incremental improvement to the bottom-up program synthesis algorithm.
- Authors have considered the efficiency issues of adding machine learning model into the bottom-up search and came up with optimizations that result in model cost being lower than efficiency gains.
- Authors figured out how to train the NN for their task absent actual training dataset.
- Algorithm outperforms baseline approaches.
- All sections of work are well presented, understandable, and easy to follow. The paper was a pleasure to read.

Concerns:
- My concerns can be summarized in short question: are contributions significant enough?
- In experiments, the algorithm barely outperforms a baseline with heuristics only. This is the case for both authors' benchmark set and SyGuS benchmark set.
- Using property signatures is not new. The novel part is adding NN to prioritize search based on property signatures. I would say that the idea to use NN for this is not significantly novel. There is some novelty in setting up training dataset as I observed above.

Possible mitigation of concerns:
- Choose more complicated problem (benchmark) set. This may show where baseline approaches do not work well. Since authors created first benchmark set themselves, I am surprised this was not done already.
- Another variation would be a useful problem set that is not more complicated, but where for some reason baseline(s) do not work.
- Provide results for a baseline that uses property signatures without NN. This is a minor mitigation: it is possible that property signatures without NN will not outperform the baseline with heuristics. It still would be interesting to see the comparison.

In summary: I am not convinced that contribution is significant enough for acceptance.

Reproducibility: It would be nice if authors presented actual NN architecture used for ease of reproducibility.

---

> ### Author Response · Authors · 2020-11-20
> **Response to Reviewer 4**
>
> Thanks very much for the review. Responses inline below:
>
> > In experiments, the algorithm barely outperforms a baseline with heuristics only.
>
> We would like to push back on this a bit.
>
> First, we think the improvement is reasonably large. The improvement in our benchmarks for "Successes by Time Elapsed" of "using model and heuristics" over "using heuristics only" is over 10% (32 / 29). It's true that the improvement is closer to 4% for SyGuS, but this leaves out a crucial point: we didn't retrain the neural network for SyGuS problems! Furthermore, the benchmark tasks have widely varying difficulty, and since the search space grows exponentially with the solution size, solving even a few more problems shows a significant improvement in the synthesizer’s ability to scale to find larger solutions.
>
> Second, we put a lot of effort into crafting those strong heuristics. Many other synthesis papers do not bother comparing their method to well-engineered heuristics. If we had excluded the heuristics, our improvements would have been 31 / 26 (19%) for our 38 benchmarks and 75 / 65 (15%) for SyGuS. In our opinion, authors should not be penalized for engineering strong baselines for comparison.
>
> Third, the heuristics are only possible to implement because of the way we set up the search (bottom-up where we can evaluate every intermediate expression to obtain concrete results).
> We could reasonably claim that our search technique coupled with the heuristics is itself a new contribution of this paper.
>
> > I would say that the idea to use NN for this is not significantly novel
>
> We have made the specific contribution of noticing that when you do bottom-up search in which all intermediate values can be evaluated, you can train a model to predict whether those values will appear in a solution. This is not something that has been done before. Moreover, we have actually made it efficient enough to yield a wall-clock speedup. We believe it is important to recognize empirical contributions that demonstrate the practical utility of new techniques.

---

> > ### Comment · AnonReviewer4 · 2020-11-20
> > **Response to authors**
> >
> > Regarding your pushback, I would like to push back too: model without heuristics only outperforms heuristics 31/29 on your benchmarks and underperforms heuristics on SyGuS. So if model by itself is a major contribution, then the performance is not great.
> > If you believe that you can "claim that our search technique coupled with the heuristics is itself a new contribution of this paper", then you should do so. That claim - in addition to your existing claims - would strengthen the paper. Especially, since ultimately it's the model+heuristics which performs best.
> >
> > Regarding "we didn't retrain the neural network for SyGuS problems" - If you have to retrain the neural network for any new set of problems/benchmarks, then there is a risk that it is overfitted. I realize that you may have to retrain for a substantially different domain, but there's a line than needs to be treaded carefully. I think with the training challenges you described, it may be difficult to analyze potential overfitting, so let's leave it as it is.
> >
> > Thank you.

---

### Official Review · AnonReviewer3 · 2020-10-28
**Review of BUSTLE - a program synthesis technique using program signatures (fusing formal methods with machine learning)**

**Rating:** 9
**Confidence:** 5

**Review:**

Note about NeurIPS ’20 version:

I was on the NeurIPS ’20 program committee and I was assigned an earlier version of this paper. As such, I’m deeply familiar with it. In its NeurIPS form, I felt it wasn’t ready for tier-1 publication. However, my concerns were principally around the lack of experimental results. I strongly supported of the ideas presented in the paper, but I fought to ensure it wasn’t accepted to NeurIPS because I felt it would have been a disservice to both NeurIPS and the authors, resulting in a rather mediocre paper that could have been exceptional if the proper experiments were run. I attempted to make this clear in my review and encouraged the authors to address this weakness.

It appears the authors have addressed my primary concern. This version of the paper resolves my critical reservation of weak empirical results – the results, I believe, are now satisfactory of tier-1 publication, which include four different synthesis systems and over 100 different synthesized programs of two different domains. As such, I now support its acceptance at ICLR.

Summary:

This paper appears to be the first work to use program signatures (ICLR ’20, Odena & Sutton) for program synthesis. The high-level concepts the authors present, as I understand them, is that by using program signatures for program synthesis, they can more closely replicate the process of program synthesis the way programmers develop programs. That is, by breaking one large program into many small sub-programs. Once enough of these small sub-programs have been generated, they can be composed together to solve the larger program – the actual goal.

Assuming this hypothesis holds, the end result might be that such an approach would result in a program synthesizer that could generate both (i) more correct programs, (ii) faster than prior systems. For their experimental evaluation, this hypothesis seems to hold and BUSTLE does in fact generate more correct program, more quickly than prior state-of-the-art. The authors compare BUSTLE to three other variant systems: a baseline (less sophisticated BUSTLE system), RobustFill (ICML 2017), and DeepCoder (ICLR 2017). RobustFill and DeepCoder have demonstrated state-of-the-art performance, historically, so I believe these comparisons are sufficient for this paper’s acceptance.

Overall, I think the paper provides a truly novel approach to program synthesis with its fusion of program signatures. I am admittedly biased in favor of program signatures, because I believe the future of machine programming / software 2.0 / neural programming / program synthesis with both stochastic and deterministic approaches (whatever we want to call it) is going to be heavily reliant on our ability to lift concepts from code (the “what”) which is notably more challenging than lifting the implementation (the “how”). This is because the “what” tends to not necessarily be obvious from the code, whereas the “how” almost always is – it’s the implementation. With that in mind, this paper presents what I believe is the first demonstrable evidence that program signatures can be used in this fashion. I suspect this is just the beginning of exploration with program signatures – I expect a flurry of follow-up research to emerge that uses them.

When taken holistically, I strongly support accepting this paper, but I do have some minor nits I’d like the authors to address.

Minor suggestions:

1. There appears to be no system diagram of BUSTLE. While an expert in the space of program synthesis and property signatures can likely understand what is going on, non-experts I think will really struggle without some kind of visual diagram showing how BUSTLE works. It should be relatively easy to add this diagram to the paper and I believe it will make the paper more widely accessible. If only one of my recommendations is addressed by the authors, I would request it be this one.

2. There appears to be multiple locations where the authors seem to deem neural network inference is “too slow” without qualification. I think this is a mistake and is a bit of a turn off and it’s a bit of a confusing one given that BUSTLE uses neural network inference. Yes, I do agree with the authors that inference with large neural networks could make the problem of program synthesis slower, but I don’t believe this is a universal truth. I think it’s proportional to the computational complexity of the neural network. I would request the authors find all such “inference is too slow” cases in the paper and properly qualify them. I suspect this will encourage future work to consider other neural network architectures that may be competitive or even outperform BUSTLE.

3. Can you label the different variants of BUSTLE from something like “Using model and heuristics” to just BUSTLE, “Using model only” to BUSTLE (model only), etc. Right now it’s a bit confusing at first glance on Figure 2 to see which is the full BUSTLE system because there isn’t actually any legend item that is called “BUSTLE”. Should be an easy fix and will likely make the figure easier for the audience to understand.

4. Can you please drop the word “very” from the paper everywhere it appears? I do not have a mathematical representation of what that word means (nor does anybody I think) and, as such, I believe it introduces unnecessary ambiguity and also wastes paper space.

5. I couldn’t tell how the BUSTLE training time was factored into the analysis. Can you find a way to explain that more clearly? I realize that it’s a potential one-time only penalty, but it doesn’t come for free (to my understanding) while some traditional program synthesis systems using formal methods can simply generate programs without any learning overhead. I think this needs to be captured somewhere so people don’t forget about this cost.

6. I got a little confused by the comments about removing restrictions of Concat() in the second paragraph in section 2.2. Can you try to explain that more clearly?

7. Can you provide some intuition on the rationale behind keeping “100 positive and 100 negative values” as explained in the last sentence in section 3.1?

8. Can you double check to ensure all of your acronyms are fully spelled out first? I’m familiar with all of them, but others might not. I don’t think I saw the spelled out versions of DSL, AST, JVM, etc. While these terms are generally widely known in the programming languages community, I’m not sure if the machine learning community is as deeply aware of them. Regardless, it seems to me that it’s usually a good idea to spell out all acronyms first.


Future work:

Do you really think an abstract syntax tree (AST) representation is the right representation for this approach?

I’m not so sure. I recommend taking a look at the Aroma’s simplified parse tree in the paper by FAIR, Berkeley, and Riverside (OOPSLA ’19) and, more comprehensively, MISIM’s context-aware semantics structure from Intel, Georgia Tech, MIT (arxiv). I suspect a next iteration of BUSTLE using either of these structures might result in even better performance than what you’ve currently achieved with an AST representation. But, that’s just a guess. :)

---

> ### Author Response · Authors · 2020-11-20
> **Response to Reviewer 3**
>
> Thank you for your in-depth review! We are glad to hear that our expanded experiments have greatly improved the paper. We also agree that large-scale success of program synthesis likely depends on reasoning about high-level concepts, and property signatures is an important step toward that direction.
>
> We’ve made many changes to the paper, following your helpful suggestions, including adding a diagram (Figure 2 in the updated paper).
>
> > I got a little confused by the comments about removing restrictions of Concat() in the second paragraph in section 2.2.
>
> Certain prior works (e.g., FlashFill, RobustFill) impose a restriction of having Concat as the top-level operation, i.e., they only handle tasks with this property. They can then use version space algebras, dynamic programming, or other pruning strategies to exploit the property that partial programs must form substrings of the output (or, a prefix of the code solution must produce a prefix of the output). Our DSL lifts this constraint, allowing the synthesizer to handle more kinds of tasks than those previous works.
>
> > Can you provide some intuition on the rationale behind keeping “100 positive and 100 negative values” as explained in the last sentence in section 3.1?
>
> Each training point consists of random inputs, a value encountered during the search which we treat as the output O, and an “intermediate value” V that is either chosen to be a sub-expression of the output, or some other randomly-chosen value from the search that is not a sub-expression of the output. We want to construct a balanced training dataset, where half of the datapoints should be predicted as positive (V is a sub-expression of O), and half are negative (V is not a sub-expression of O). Hence, from each of 1000 synthesis runs using random inputs, we extract 100 positive data points and 100 negative data points for the training set.

---

### Official Review · AnonReviewer1 · 2020-10-28
**A well written and evaluated paper combining synthesis techniques with machine learning.**

**Rating:** 6
**Confidence:** 4

**Review:**

The paper proposes to combine bottom-up program synthesis from input/output examples with a machine learning model. The machine learning model determines for each candidate intermediate value (coming bottom-up from the inputs) if it may be considered at the next round of candidate expression generation or it will be deferred for at least K rounds (with K up to 5 in the evaluation).

Pros:

- The paper is evaluated on a good range of string manipulating programs. The presentation and the evaluation is showing the advantages of the reweighting for the number of solved programs.

- The advantages of the method are not only in the number of candidates explored, but in actual wall-time. This is a relatively rare result - many prior synthesis with machine learning tasks completely actual running time.

- The paper and its implementation address the engineering side of the work - batching the requests for effective machine learning and interestingly, the results are good even when the machine learning model was trained on random data.

Cons:

- It makes it difficult to compare based on the numbers, but it looks like the result is not competitive with existing solvers for the CyGuS PBE competitions. The paper should mention where it stands here.

- The actual contribution of the work is mostly in the implementation and combining known techniques.

In terms of writing, there are some improvements that are possible:

- The algorithm description is not self-contained. It is not completely clear what ExtractConstants does. The inputs I and outputs O should be vectors, but they are used as sets. The algorithm actually is unclear here. Does the set E[1] include the entire input vector for input examples as an element and each constant as a vector of the constants with this length (In this case it should be E[1] = {I} \cup C)?
- There is also no intuition for what concrete property signatures make sense.

---

> ### Author Response · Authors · 2020-11-20
> **Response to Reviewer 1**
>
> Thank you for your review!
>
> > It makes it difficult to compare based on the numbers, but it looks like the result is not competitive with existing solvers for the CyGuS PBE competitions. The paper should mention where it stands here.
>
> The CVC4 solver is able to solve more SyGuS tasks than other methods, but its solutions are often very long, difficult to understand, and overfit to the I/O examples. For example, CVC4 generates a solution with more than 5000 AST nodes for one of the string benchmarks, mostly consisting of a case split style program. In contrast, enumerative searches like BUSTLE prioritize shorter programs over longer ones, and thus the synthesized programs are more interpretable and more likely to generalize. We instead compare against DeepCoder and RobustFill, which are state-of-the-art synthesis systems using machine learning. Additionally, the DSL used in our paper is different and more general than the DSL used in SyGuS tasks, e.g., BUSTLE handles changing case of strings while this is not supported by SyGuS.
>
> > The actual contribution of the work is mostly in the implementation and combining known techniques.
>
> Our main contribution is applying machine learning in the loop for bottom-up synthesis, and showing that it can be done quickly enough for wall-clock speedups overall. To our knowledge, no other work has used machine learning within a bottom-up search in this manner; previous works like DeepCoder use machine learning only at the beginning and hence do not guide the search as it happens, or use an end-to-end ML approach like RobustFill. As AnonReviewer5 mentioned, the simplicity of our approach (which is novel nonetheless) is one of its strengths.
>
> > It is not completely clear what ExtractConstants does.
>
> ExtractConstants uses heuristics to select constants from a set of I/O examples, including common symbols/delimiters that appear in the example strings, and long substrings that appear multiple times in the example strings. We added this clarification to the paper.
>
> > The inputs I and outputs O should be vectors, but they are used as sets. The algorithm actually is unclear here. Does the set E[1] include the entire input vector for input examples as an element and each constant as a vector of the constants with this length (In this case it should be E[1] = {I} \cup C)?
>
> You are right, I and O are confusing in the algorithm. Suppose there are M separate input variables (such that the program we wish to synthesize is a function with arity M), and there are N different I/O examples. Every “value” in the algorithm represents a code expression and internally contains the result of that code expression when applied to each of the N examples. The output value, O, is a single value (as usual, containing the desired outputs for all N examples). The input values, I, is a set of M values, each of which contains N strings (the example input strings for that input variable). When we compare values in the algorithm (“if V = O then: return Expression(V)”), we are comparing whether all N strings match, so in this case, a code expression will be returned if it evaluates to the desired output for all N examples. With this clarification, “E[1]←I∪C” is correct -- each of the M input variables, as well as each extracted constant, becomes a value with weight 1. We’ve updated the paper to make this more clear.
>
> > There is also no intuition for what concrete property signatures make sense.
>
> Recall that the model is conditioned on two property signatures: one from the inputs to the output, and one from the intermediate value to the output. Listing 1 shows 3 example property signatures, which compare whether an input is a substring of the output, a suffix of the output, or a substring of the output when ignoring case. These are all properties that are useful for string manipulation:
>
> * If the input is a substring of the output, then the output is likely to be constructed by concatenating other strings to the input.
> * If the input is only a substring of the output when ignoring case, then the desired program likely includes some case-changing functionality.
> * If the intermediate value is a substring of the output, then it is more likely to be useful than another value that has no relation to the output.
>
> Also, listings 5-8 contain all of the properties we use.

---

> > ### Comment · AnonReviewer1 · 2020-11-24
> > **Clarification**
> >
> > Thank you for your response.
> >
> > I believe the algorithm is much more clear now.
> >
> > I would also appreaciate if the authors include the results for the CVC4 (and Z3) solver in the next revision of the paper as it should not be that the readers need to find it in the comments of reviews and the paper did some selection of tasks, so the results are readily available elsewhere.

---

### Official Review · AnonReviewer5 · 2020-11-02
**Compelling approach to bottom-up program synthesis**

**Rating:** 8
**Confidence:** 3

**Review:**

# Summary

The paper proposes an approach to program synthesis which is done in a bottom-up fashion.
In order to guide the search more effectively the bottom-up search algorithm is accompanied with a model predicting whether particular sub-expressions of a program are promising directions.
As the system for program synthesis needs to be real-time capable, the proposed approach heavily relies on property signatures as introduced in (Odena & Sutton, 2020) for featurizing program inputs.
The here proposed system called BUSTLE is shown to perform favorably to a set of baselines including state-of-the-art approaches to program synthesis.

# Resons for Score
Generally speaking, I really enjoyed the reading of the paper. The paper is well structured and written as well as easy to follow.
Everything is explained in sufficient detail, i.e., thorough but concise.
The contribution of the paper is significant and the approach is technically sound.

# Pros

Although the methods might not be feasible to be used in practice yet, it performs reasonably well with only a few examples for describing the desired behavior of the programs to be synthesized.
However, the programs that are synthesized are still quite smallish. It would be interesting if the authors could also highlight some future directions in order to make a step towards more complex scenarios.

The paper provides compelling experiments which are set up in a thoughtful and rational manner rather than throwing various methods on some benchmarks.
The experiments are conducted in a very systematic way, facilitating insights into the performance of the proposed method as well as how it compares to the baselines.
However, a taxonomy of problems (e.g. how to quantify problem severity etc.) would maybe help to better spot differences in performance.

The method itself is quite simple. The authors almost seem to apologize or vindicate. I rather consider it an advantage.

The proposed approach is well placed in the context of the existing literature.

# Cons

Some information about the experiment setup is missing: Are all benchmark tasks of one set (either on the proposed set or SyGuS) run in a row?
Is the order kept consistent? What hardware is used for conducting the experiments?

# Questions during rebuttal period

- Out of curiosity: If the order is the same for all the benchmarks for all methods, it seems to be the case the method being fastest to find a solution for a benchmark task varies from task to task,
and the here proposed method does not always seem to be the fastest for any instance. Given the demanding real-time setting, do you think an algorithm selection approach would work here, choosing a method
on a per task basis?



# Typos
- p.3: "during the search, [...]" => During the search
- p.6: "fixed length representation" => fixed-length representation (this is occurring at least twice)
- p.8: "This provides further evidence the our model [...]" => This provides further evidence that our model [...] (occurs also in the appendix)
- p.8 last but one paragraph in Section 5: "[...], while like BUSTLE, uses values produced by intermediate programs [...]" => The comma after BUSTLE is kind of irritating. Please, consider removing it or adding another comma before "like".

---

> ### Author Response · Authors · 2020-11-20
> **Response to Reviewer 5**
>
> Thank you for your review! In particular we agree that BUSTLE’s simplicity is a strength of the method. In our updated paper, we have fixed all the typos you pointed out.
>
> > Out of curiosity: If the order is the same for all the benchmarks for all methods, it seems to be the case the method being fastest to find a solution for a benchmark task varies from task to task, and the here proposed method does not always seem to be the fastest for any instance. Given the demanding real-time setting, do you think an algorithm selection approach would work here, choosing a method on a per task basis?
>
> The benchmarks are run in a consistent order in our experiments, but actually the order does not matter for the plots. We ran each method for the full timeout of 30 seconds on _each_ benchmark task, and the plot shows the number of tasks where the method succeeded (Y-axis) before a specific amount of time had elapsed for that task (X-axis), and similarly for the number-of-expressions plots. Your observation is still correct though -- no method is consistently the fastest for all tasks. We believe that an algorithm selection approach could lead to improvements, but doing so would require different training algorithms (possibly reinforcement learning). Similarly we might be able to improve the BUSTLE approach as well, training the model by using real data from the synthesizer instead of synthetic data.

---

> > ### Comment · AnonReviewer5 · 2020-11-23
> > **Re: Response**
> >
> > Thank you very much for the clarification and for your detailed response.

---

### Comment · AnonReviewer4 · 2020-11-15
**I am fine with accept**

Although I stand behind my thoughts and comments in my review, I am fine with "accept" decision based on the arguments of other reviewers. Hopefully my input will still be useful for authors.

---

> ### Author Response · Authors · 2020-11-20
> **Review score**
>
> You're obviously free not to do this, of course, but we wonder if -- in light of this comment -- you'd consider updating your score to 6 instead of 5? We feel that this more closely matches a position of "I am fine with accept" than does 5, though of course we realize that different people may have different interpretations of the numbers. We have indeed found your other comments useful - see our response below.

---

### Decision · Program_Chairs · 2021-01-07
**Final Decision**

**Decision:**

Accept (Spotlight)

**Comment:**

The reviewers have supported the acceptance of this paper (R3 and R5 were particularly excited) so I recommend to accept this paper.